# Recent Trends in Dietary Habits of the Italian Population: Potential Impact on Health and the Environment

**DOI:** 10.3390/nu13020476

**Published:** 2021-01-31

**Authors:** Marilena Vitale, Annalisa Giosuè, Olga Vaccaro, Gabriele Riccardi

**Affiliations:** 1Department of Clinical Medicine and Surgery, Federico II University, 80131 Naples, Italy; marilena.vitale@unina.it (M.V.); annalisa.giosue@gmail.com (A.G.); riccardi@unina.it (G.R.); 2Department of Pharmacy, Federico II University, 80131 Naples, Italy

**Keywords:** food choices, food balance sheets, diet quality, environmental impact, human nutrition, ecological impact, sustainability, EAT–Lancet commission

## Abstract

Population growth, globalization, urbanization, and economic pressures are causing changes in food consumption all over the world. The study’s aims are (1) to evaluate trends in food habits in Italy to highlight deviations from the traditional Mediterranean diet, (2) to analyze the features of the present Italian diet that should be modified to meet evidence-based global scientific targets for a healthy and sustainable diet proposed by the EAT–Lancet Commission. Trends in food availability for human consumption during the period 2000–2017 were assessed using the food balance sheets (FBSs). Greenhouse gas (GHG) emission was estimated according to life cycle assessment (LCA) analyses. During the study period, the availability of animal fat and beef meat greatly declined (−58% and −32%, respectively), followed by fruit, potatoes, vegetables, milk, and non-tropical oils (−20%, −15%, −13%, −14%, and −11%, respectively). A substantial increase has occurred for tropical oils, fish, and nuts (+156, +26%, and +21%, respectively). In order to meet the targets of consumption proposed by the EAT–Lancet Commission, the consumption of legumes and nuts should be almost doubled, whereas the consumption of meat, eggs, dairy products, animal fat, tropical oils, and sugars should be reduced by proportions ranging from 60% to 90%. If implemented, these changes would reduce the diet-related greenhouse gas emission by nearly 50%. In conclusion, these data call for nutritional education programs and interventions on the food system aimed at promoting a healthier and more environmentally sustainable diet. To this end, the availability and affordability of products with a better impact on human health and the environment should be promoted.

## 1. Introduction

A large body of evidence highlights dietary choices as lifestyle features strongly associated with chronic non-communicable diseases (NCDs), such as cancer, cardiovascular disease, and diabetes, that represent the main cause of premature death and disability [1,2]. The traditional Mediterranean diet is a balanced model of healthy eating deeply rooted in a rich cultural and gastronomic background. Because of its distinctiveness in terms of practical knowledge and traditions and its strong cultural foundations, the Mediterranean diet has been recognized by UNESCO as an “Intangible Cultural Heritage of Humanity.” 

The health benefits of the traditional Mediterranean diet have been well documented over the years. Several studies have consistently shown a positive association between a diet resembling the Mediterranean model, a longer lifespan, and a lower risk of diabetes, cardiovascular disease, and cancer; more recent evidence suggests that also cognitive decline and chronic digestive diseases occur less frequently. These findings have been consistently reproduced in different countries and in various ethnic groups [3,4,5,6,7,8,9].

In addition, the Mediterranean diet represents a potentially useful means to achieve the Agenda 2030 Sustainable Development Goals because it can help reduce the environmental impact of the diet [10]. The relevance of the food choices in relation to ecological sustainability is gaining increasing support in the scientific literature. Food production and distribution contribute at least one-third of the greenhouse gas (GHG) emissions, thus playing a significant role in climate change. However, not all foods have the same ecological impact; in general, foods of animal origin have a greater impact on environmental degradation than those of vegetable origin [11]. 

Italy has been traditionally linked to the Mediterranean diet since the pioneering work of Keys et al. [12]. The traditional Italian cuisine is typically based on large quantities of vegetables, fruit, cereals, legumes, nuts, and limited amounts of animal products. In addition, it has another important feature—the use of olive oil, rather than animal fat, for cooking [13]. However, population growth, globalization, urbanization, and economic pressures are causing changes in the food systems, eating habits, and food consumption patterns towards a globalized approach, with undesired effects on people’s health and the environment [14]; in this process, Italy is no exception. Providing a growing global population with healthy and environmentally more sustainable food options is an urgent challenge. To this aim, the “EAT–Lancet Commission on healthy diets from sustainable food systems” has produced a reference diet with indications for recommendable ranges of intake [14]. This commission brings together 19 commissioners from 16 countries, experts in various fields of human health, to develop global scientific targets based on the best evidence available for healthy diets and sustainable food production.

Against this background, the study’s aims are (1) to evaluate trends in food habits in Italy to highlight any deviations from the traditional Mediterranean diet and (2) to analyze the features of the present-day Italian diet that should be modified to meet the evidence-based global scientific targets for a healthy and sustainable diet proposed by the EAT–Lancet Commission.

## 2. Materials and Methods

### 2.1. Food Balance Sheets (FBSs)

The evolution of food availability over time in Italy has been assessed by the food balance sheets (FBSs), the statistical databases of the Food and Agriculture Organization (FAO) of the United Nations, which provide a comprehensive picture of the food available for consumption in each country, during a specified reference time period [15]. The food supply during each year is estimated as the sum of the total quantity of foodstuffs produced (including production, imports, and stock changes) minus exports, food use other than human feed (including animal feed, seeds for agricultural use, etc.), and food losses during food transport, storage, and processing. The per capita food supply available for human consumption in Italy is then obtained by dividing the respective quantity of each food group by the number of people living in the country in the same time period. Because FBS data do not consider food waste occurring with food distribution and consumption at the household level, for each food category we have calculated the amount of food available for human consumption by subtracting from the per capita food supply obtained from FBSs, the presumable waste, as calculated by FAO [15,16], following the formula of the amount of per capita food available for human consumption (FBS data) minus presumable waste (FAO). The reported estimate of food loss and waste has remained stable during the years. The data reported in the tables and figures pertain to “food available for human consumption”, and the terms “food consumption,” “food intake,” or “food supply” should be read as “food available for human consumption.” Raw data are given in the Appendix A. 

Data were extracted for the following food groups—fruits, vegetables, legumes, nuts, potatoes, cereals, beef and pork meat, poultry, fish, eggs, milk, animal fat, tropical oils (coconut, palm, and palm kernel), non-tropical oils (maize, olive, sunflower, rapeseed, peanut, and soybean), and sugars. Subsequently, the average per capita supply of macronutrients (i.e., energy, protein, and fats). as reported by the FBSs, was used in the analyses; carbohydrate content was estimated using the following formula: 

Carbohydrate content in grams = total energy − (energy from fat + energy from protein + energy from alcohol) /4 (the energy value provided by 1 gram of carbohydrates).

Consumption trends were evaluated starting from the beginning of this century up to the year 2017, which represents the most updated information available. Because some difficulties are encountered when estimating trade, production, and stock changes on an annual scale, we reported the mean and range of three-year periods to minimize estimation errors; the average figures relative to the last available observation (i.e., years 2015–2017) has been compared with the average intake recommended by the EAT–Lancet Commission [14] to highlight the desirable changes in consumption necessary to meet the recommendations. 

### 2.2. Green House Gas (GHG) Emission

GHG covers CO_2_ emissions from fossil fuels, CH_4_ released during cattle rearing and the cultivation of certain crops, and N_2_O released from fertilizers, manure, and plowing of grassland. It is expressed as Kg CO_2_-equivalents (CO_2_e) calculated by adding CO_2_eq, N_2_O, and CO_2_eq CH_4_ to CO_2_ emissions. These metrics are estimated according to life cycle assessment (LCA) analyses. LCA is an internationally recognized method employed to estimate the impact of resource use at all stages of a product’s lifespan [17]. For most food groups, we used the GHG data provided by the study of Naja et al. [11], which was conducted in the Mediterranean area (Lebanon) with climate and food choices not dissimilar from Italy’s. In order to obtain GHG values expressed in a conventional way, i.e., Kg/capita/year–data on food supplies expressed as g/capita/day was multiplied by 365/1000 and then multiplied again by the corresponding values of GHG per Kg food. When a food group included multiple items, CO_2_ was calculated separately for each item based on the amount available for consumption according to FBS, and then summed-up.

## 3. Results

Data on the quantity of each food group available for consumption over time—corrected for waste—are reported in Table 1; they represent the average of three-year periods. Information on the crude yearly data without waste correction is given in Appendix A.

In the time period 2000–2017, the amount available for consumption has remained stable (i.e., <5% observed change) for cereals, legumes, pork meat, poultry, eggs, and sugars. A relevant decline is observed for animal fat and beef meat (−58% and −32%, respectively), with negative trends starting from the years 2009–2011. Fruit, potatoes, and vegetable supplies have also progressively declined in comparison with the years 2000–2003, albeit to a lesser extent (i.e., −20%, −15%, and 13%, respectively); these trends began in the first decade of the century for potatoes and vegetables, and more recently, in the years 2012–2014, for fruit. During the study period, the consumption of milk (including dairy foods) and non-tropical oils have also declined substantially (−14% and −11%, respectively) (Table 1, Figure 1).

The food groups with a marked increase in their supply over the observation period were tropical oils, fish, and nuts (+156%, +26%, and +21%, respectively), although with a different pattern—for nuts, a steady increase occurred at the beginning of the century, remaining fairly constant ever since; conversely, for tropical oils and fish, there has been a progressive rise over time (Table 1, Figure 1).

The pattern of the energy and nutrient composition of the diet, which is calculated by summing up the macronutrient content of all foods available for consumption during each of the three-year periods, is reported in Table 2. 

The data clearly show that no important changes occurred during the observation period in terms of total energy, fat, proteins, and carbohydrates, with the only exception of energy and fat from foods of animal origin, which decreased by 10% and 9%, respectively. Declines in energy and fat of animal origin have started already at the beginning of the century, expanding gradually up to the last available observations.

Table 3 reports quantities of foods available for consumption at the last available observation (2015–2017) as compared to the targets proposed by the EAT–Lancet Commission for healthy and environmentally sustainable dietary choices. 

To meet the EAT–Lancet targets, the consumption of fruits, vegetables, legumes, and nuts, although still within the recommended ranges, should be substantially increased in the current Italian diet; conversely, the consumption of beef and pork meat eggs, milk and dairy products, animal fat, tropical oils, and sugars should be reduced. The most relevant decreases should concern beef and pork meat, animal fat, and sugar, which should be reduced by amounts ranging from 60% to 90% (Table 3). The consumption of poultry, fish, and non-tropical oils, although above the targets, falls within the recommended ranges, however, and should not change.

In Table 4 we compared the GHG production induced by the foods available for consumption in Italy in the years 2015–2017 with the expected GHG production under the hypothesis that food consumptions would comply with the targets recommended by the EAT–Lancet Commission. The present Italian diet is associated with a CO2 equivalent production of 1465 Kg/person/year, while the EAT–Lancet Commission diet is expected to produce 740 Kg/person/year (Table 4). 

This improvement in the environmental impact of the diet would be due largely to reduced use of animal products and represents the balance between the modest increase (+77 Kg CO_2_ equivalent/capita/year) due to the greater consumption of legumes and nuts and the large decrease due to the reduced consumption of red meat and dairies, which contribute more than 40% of the GHG (Table 4). 

## 4. Discussion

This study shows that food habits in Italy have changed considerably since the beginning of the century; however, changes in the consumption of specific food items were not uniform— some of them increased and some others declined, overall balancing each other in terms of energy supply (Table 2). This is in line with the observation that the prevalence of overweight/ obesity in Italy has remained fairly stable in this period [18].

The recent changes in dietary habits in Italy were only partially coherent with recommended targets for a healthy and sustainable diet. In particular, during the last twenty years, the consumption of beef meat, animal fat, milk, and dairy products has decreased, while the supply of nuts has increased; these changes represent an improvement of the diet in relation to its potential impact on NCD prevention and environmental sustainability. Unfortunately, other food choices have changed in the opposite direction; in fact, the supply of vegetables and fruit has dropped, and the supply of tropical oils has increased (Table 1).

Reductions in supplies of fruit and vegetables have started only recently, despite the large body of scientific evidence accumulated in the last decades on their health benefits and the indications of many influential health authorities that strongly recommend the consumption of these items [19,20,21]. The reduced amount of fruit and vegetables in the Italian diet during recent years goes in parallel with similar trends reported in other European countries [22,23]. There are no clear explanations for this observation; one possibility is the aging population—elderly people tend to consume less of these foods. Other factors may, however, be at play, including accessibility, affordability, and availability of these products that may substantially impact consumption [22,24,25,26,27]. Climate changes may also have a relevant effect.

The recent modifications in dietary habits in Italy are only partially coherent with the recommended targets for a healthy and sustainable diet proposed by the EAT-Lancet commission. Overall, the data indicate that most dietary choices of the Italian population should be modified to comply with these targets. The major desirable changes would be a further remarkable drop in the consumption of red and processed meat (75–90%), animal fat (60%), sugar (60%), and dairy products and eggs (50%). The consumption of nuts and legumes should, however, be promoted based on the available evidence on the beneficial health effects of these foods, including their relevant contribution to the protein supply [14]. 

The health benefits of greater adherence to the recommended targets are supported by a large body of literature [28]. Several prospective observational studies have convincingly shown that the incidence of NCDs, in particular, cardiovascular disease and type 2 diabetes, is associated with high intakes of red and processed meat, animal fat, and sugar—particularly soft drinks—and with low consumption of vegetables, fruit, legumes, nuts, and non-tropical vegetable oils [29]. Data on the relationships between NCDs and dairy foods and eggs are less consistent but overall indicate possible harm at very high intakes [28,30]. Moreover, the PREDIMED (Primary Prevention of Cardiovascular Disease with Mediterranean Diet) trial has shown that the implementation of a diet resembling the traditional Mediterranean diet, in many ways similar to the dietary pattern proposed by the Eat–Lancet Commission, substantially reduces the incidence of cardiovascular diseases and diabetes [31,32]. 

Regrettably, adherence to the traditional Mediterranean diet is at present rather modest in Italy, despite its strong linkage with the Italian gastronomic tradition. Population-based studies performed with the use of food frequency questionnaires document a high intake of meat and a low intake of vegetables and fruit, and legumes [33,34,35]. The reported recent trends in the Italian diet can significantly impact the health of the general population since dietary habits represent the third leading risk factor for death with a strong impact also on disability-adjusted life years (DALYs), which include years lost due to premature death and the fraction of years lived in less than full health [36]. Moving toward the dietary pattern promoted by the Eat–Lancet Commission would reduce the burden of diseases attributable to unhealthy dietary habits and would also lead to a substantial reduction of CO2 emissions. A yearly saving of 725 Kg CO_2_ emission per person (Table 4)—roughly corresponding to the GHG emission of a car covering 4000 Km—would be obtained if all the proposed dietary targets were met [37]. These findings are qualitatively in line with those of several studies showing that the traditional Mediterranean diet, predominantly based on foods of vegetal origin, has a lower ecological impact and, in particular, a low GHG emission compared to present-day dietary habits in many western countries [38] and in the Mediterranean region [37,38,39,40,41].

As for the impact of each dietary component on GHG, our study shows that the sole reduction of red and processed meat consumption in the habitual Italian diet would cut the per capita production of CO_2_ equivalent by as much as 31% and contribute to more than 50% to the GHG savings obtained by following the EAT–Lancet Commission recommendations. However, considering that too sharp reductions of red and processed meat intake are rather difficult to achieve in the short term, it is worth underlining that, for the time being, even less dramatic changes in their consumption in Italy would be relevant and should therefore be promoted [42]. 

Our study has strengths and limitations. The study strengths are the assessment of the trends in food choices up to recent years (2000–2017) and the evaluation of the nutrition transition in a country deeply rooted in the traditional Mediterranean diet. Moreover, it is also worth underlining that food choices in this study have been assessed not only in relation to their nutritional relevance but also for their environmental sustainability. In this respect, we have performed quantitative evaluations of the ecological impact of the current Italian diet and of the improvements that could be achieved were the EAT -Lancet targets met. 

As for limitations, we have to acknowledge that, because in Italy no regular assessment of food habits at the population level is performed, we had to rely on foods available for consumption (food balance sheets, FBSs) rather than on direct measurements of dietary habits. This was the only available tool to look at dietary changes over time. However, in the interpretation of FBS data, it is important to consider that they exceed by a variable extent the real food intake since they include also wasted foods; in order to overcome, at least partially, this source of inaccuracy, we have corrected the crude data by the rate of waste for each food according to FAO [16]. Moreover, in this study food supplies were used either to assess temporal trends or to evaluate the relative contribution of each food group to the total energy content of the diet; within this framework, a global overestimate of food consumption would not represent a source of systematic bias.

## 5. Conclusions

In conclusion, recent changes in dietary choices of the Italian population are heterogeneous in terms of their impact on health and the environment since some of them (less red meat and animal fat and higher nut consumption) are beneficial, while others (vegetables and fruit reduction) are harmful. Moreover, consumption of legumes, pork meat, and sugars have been fairly stable over time, while more legumes and less pork meat and sugar would have been more appropriate. Overall, despite the strong links between the traditional Mediterranean diet and the Italian cuisine, present-day eating habits of the Italian population are not fully coherent with a healthy and sustainable diet. 

The information provided by our study can represent a useful background to promote nutritional education programs together with interventions on the food system aiming at improving the availability and affordability of products with a better impact on human health and the environment. Moreover, the data may help to understand which features of the traditional Mediterranean diet are more difficult to reconcile with people’s lifestyles in a globalized world, thus facilitating tailored measures for the promotion of the Mediterranean diet in different cultural backgrounds.

## Figures and Tables

**Figure 1 nutrients-13-00476-f001:**
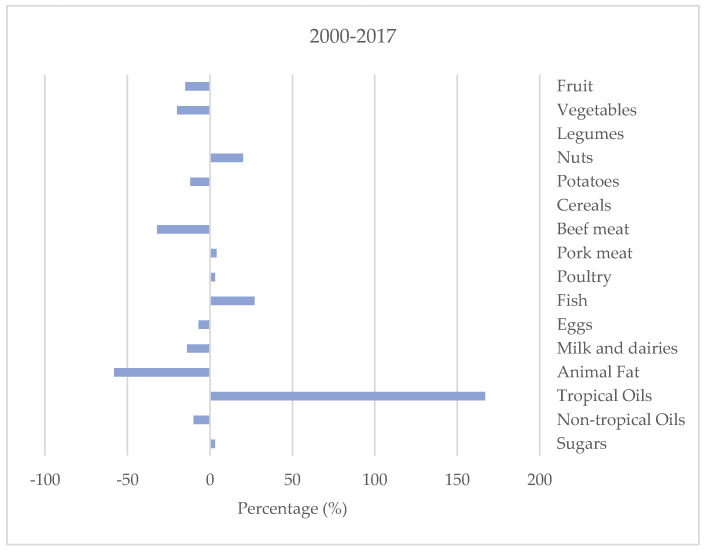
Changes in the amounts of foods available for consumption in Italy (Food Balance Sheets) from 2000 to 2017.

**Table 1 nutrients-13-00476-t001:** Food supply in Italy over time according to the food balance sheets (FBSs).

Food Supply (g/per capita/day) by Time							
	Food Loss and Waste %	2000–2002	2003–2005	2006–2008	2009–2011	2012–2014	2015–2017
Fruit	45	213 (201–221)	222 (202–233)	240 (234–243)	237 (217–263)	196 (182–210)	181 (171–195)
Vegetables	45	256 (222–301)	283 (267–301)	241 (241–246)	240 (222–269)	202 (197–208)	205 (202–208)
Legumes	20	12 (12–12)	12 (12–12)	12 (12–12)	12 (11–12)	12 (11–12)	12 (12–12)
Nuts	20	15 (14–15)	15 (14–16)	17 (16–19)	17 (16–19)	16 (15–17)	18 (16–19)
Potatoes	50	57 (55–59)	54 (51–57)	53 (51–54)	54 (53–55)	51 (49–53)	50 (49–50)
Cereals	35	289 (288–290)	283 (279–287)	279 (278–279)	278 (275–282)	280 (279–282)	288 (287–288)
Beef meat	20	53 (52–55)	52 (52–54)	52 (50–53)	50 (47–52)	43 (41–45)	36 (34–37)
Pork meat	20	85 (82–87)	87 (85–88)	88 (85–90)	89 (87–92)	85 (81–88)	88 (84–96)
Poultry	20	40 (39–42)	34 (33–34)	34 (30–37)	39 (38–40)	41 (41–42)	41 (41–41)
Fish	30	45 (43–47)	47 (46–48)	49 (48–50)	50 (48–51)	51 (49–54)	57 (56–57)
Eggs	10	30 (29–31)	29 (29–29)	28 (27–29)	29 (28–31)	31 (29–33)	28 (28–29)
Milk and dairies	10	664 (662–667)	653 (645–668)	654 (637–664)	638 (635–643)	619 (609–630)	571 (556–595)
Animal Fat	10	33 (33–34)	32 (31–33)	34 (33–34)	31 (31–32)	26 (15–33)	14 (14–14)
Tropical Oils	20	3 (3–3)	3 (3–4)	4 (4–5)	7 (6–8)	9 (8–9)	8 (8–8)
Non-tropical oils	20	51 (51–52)	50 (48–53)	53 (52–54)	51 (49–52)	48 (45–51)	46 (45–47)
Sugars	10	77 (76–79)	78 (77–79)	73 (73–74)	74 (73–76)	79 (78–81)	79 (77–80)

Data are corrected for food loss and waste, calculated according to the Food and Agriculture Organization (FAO) (16) and represent the means and ranges () of three-year periods.

**Table 2 nutrients-13-00476-t002:** Average per capita supply of Energy (Kcal/per capita/day) and Macronutrient Supply (g/per capita/day) by time.

	2000–2002	2003–2005	2006–2008	2009–2011	2012–2014	2015–2017
**Total Calories**	2583 (2574–2590)	2538 (2518–2550)	2558 (2553–2564)	2550 (2547–2554)	2528 (2470–2558)	2490 (2482–2504)
Animal Food Sources	800 (791–811)	775 (766–786)	783 (777–789)	769 (762–781)	764 (742–786)	732 (717–743)
Vegetable Food Sources	1782 (1780–1784)	1763 (1732–1784)	1775 (1764–1782)	1781 (1773–1785)	1764 (1729–1792)	1757 (1746–1765)
**Total Proteins**	79 (79–79)	78 (78–78)	77 (77–77)	78 (77–78)	76 (76–76)	76 (75–76)
Animal Food Sources	47 (47–47)	46 (45–46)	46 (46–46)	46 (46–47)	45 (45–46)	45 (44–45)
Vegetable Food Sources	32 (32–32)	32 (32–32)	31 (31–32)	31 (31–32)	31 (30–31)	31 (31–31)
**Total Fat**	124 (123–125)	121 (119–122)	126 (126–126)	125 (123–127)	123 (117–127)	118 (117–120)
Animal Food Sources	62 (62–63)	60 (59–61)	61 (61–61)	60 (59–60)	59 (56–62)	56 (54–56)
Vegetable Food Sources	62 (62–62)	61 (58–63)	65 (64–65)	66 (64–68)	64 (61–66)	62 (61–63)
**Total Carbohydrates**	307 (307–307)	303 (300–307)	298 (297–299)	297 (293–302)	298 (295–301)	300 (299–301)
Starch	188 (187–188)	183 (181–186)	183 (182–183)	183 (178–186)	183 (180–187)	184 (184–185)
Sugars	119 (119–119)	121 (119–123)	115 (113–117)	114 (112–118)	114 (109–119)	116 (115–117)

Data represent the means and ranges () of three-year periods.

**Table 3 nutrients-13-00476-t003:** Amounts of foods available for consumption (average and ranges) in Italy at the last available observation (years 2015–2017) compared to the targets and ranges proposed by the EAT–Lancet Commission.

	Foods available for Consumption- Years 2015–2017 (g/per capita/day)	EAT-Lancet Targets (g/per capita/day)	Changes Needed to Meet the Targets	Changes Needed to Meet Target Ranges
Fruit	181 (171–195)	200 (100–300)	⇑	⇔
Vegetables	205 (202–208)	300 (200–600)	⇑	⇔
Legumes	12 (12–12)	50 (0–100)	⇑⇑	⇔ *
Nuts	18 (16–19)	50 (0–75)	⇑⇑	⇔ *
Potatoes	50 (49–50)	50 (0–100)	⇔	⇔
Cereals	288 (287–288)	232 (according to energy needs)	⇔	⇔
Beef meat	36 (34–37)	7 (0–14)	⇓⇓	⇓⇓
Pork meat	88 (84–96)	7 (0–14)	⇓⇓	⇓⇓
Poultry	41 (41–41)	29 (0–58)	⇓	⇔
Fish	57 (56–57)	28 (0–100)	⇓	⇔
Eggs	28 (28–29)	13 (0–25)	⇓	⇓
Milk and dairies	571 (556–595)	250 (0–500)	⇓⇓	⇓
Animal Fat	14 (14–14)	5 (0–5)	⇓⇓	⇓⇓
Tropical Oils	8 (8–8)	7 (0–7)	⇓	⇓
Non-tropical Oils	46 (45–47)	40 (20–80)	⇓	⇔
Sugars	79 (77–80)	31 (0–31)	⇓⇓	⇓⇓

* The consumption of these foods is interchangeable; their recommended total daily consumption should be at least 50 g. ⇑ = to increase, ⇓ = to reduce, ⇔ = no variation

**Table 4 nutrients-13-00476-t004:** Annual greenhouse gas (GHG) emissions related to the amounts of foods available for consumption in Italy at the last available observation (years 2015–2017) as compared to that induced if food consumption would adhere to the targets of the Eat–Lancet Commission.

GHG (Kg CO_2_ eq/capita/year)				
	Italy (Years 2015–17)	EAT-Lancet Targets	Difference	
Fruit	28	31	+3	+77
Vegetables	111	162	+51	
Legumes	5	23	+18	
Nuts	3	8	+5	
Potatoes	2	2	0	
Cereals	141	114	−27	−802
Beef meat	302	59	−243	
Pork meat	224	18	−206	
Poultry	79	56	−23	
Fish	136	66	−70	
Eggs	40	19	−21	
Milk and dairies	312	137	−175	
Animal fat	30	11	−19	
Tropical oils	5	3	−2	
Non-tropical oils	26	23	−3	
Sugars	21	8	−13	
Total	1465	740	−725	

## Data Availability

The data presented in this study are openly available online: http://www.fao.org/faostat/en/#data.

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
