# Peer review of "Recent Trends in Dietary Habits of the Italian Population: Potential Impact on Health and the Environment"

_nutrients, 2021, doi:10.3390/nu13020476_

Round 1

Reviewer 1 Report

This report analyzes trends of food availability in Italy during the period 2000-2017 using data from food balance sheets (FBSs), changes during this period, and the concordance of actual amounts of consumption of different foods with the targets proposed by the EAT–Lancet Commission. Then authors estimate the changes needed towards a healthy diet and the savings in greenhouse gas emissions if these changes are implemented. The results suggest that Italians much deviate from the EAT-Lancet Commission targets and that they need to increase legume and nut consumption and reduce meat, fish, eggs, dairy products, animal fat, tropical and non-tropical oils, and sugars by specified amounts, which would lead to a 50% decrease in the environmental impact of the diet as assessed by standard greenhouse gas emissions attributed to these foods.

Given the (acknowledged) limitations of the FBS technique to estimate actual food consumption, even after accounting for food losses, the results and recommendations thereof should be taken with caution. The EAT–Lancet Commission targets for the desirable amounts of foods to be consumed, encompassing both human health and environmental sustainability, are also approximate, and the healthy reference amounts are shown with a “possible” range. In the present report only the median desirable intake in g/day is considered, which leads to many deviations in the actual Italian diet (Table 3). Authors should consider the following points to be addressed:

  1. The food data in Tables and Figure are only “averages” (Means? Medians?) and no index of variability is provided. In the same way that the EAT–Lancet Commission shows ranges of intake, authors should provide some measure of variability (such as IQ ranges, SDs or 95% CIs). Then, if the desirable EAT–Lancet Commission figure for a given food falls within the range of consumption in Italy, it can be considered that no major deviations exists.
  2. Another way of looking into this matter is consider if the Italian food data fall within the “possible” range of the EAT–Lancet Commission. Then, only meats, poultry, eggs, animal fat, tropical oils and sugars would deserve a recommendation for reduction, which would fit with most data derived from diet questionnaires in Western populations. I suggest adding another column to Table 3 with changes needed to meet target “ranges”.
  3. For two particularly healthy food items, fish and non-tropical oils (which means olive oil in Italy), it is preposterous to suggest that Italians should reduce their consumption because it is slightly above the EAT–Lancet Commission figures. Fish is encouraged in all healthy diets, particularly in the Mediterranean diet, and meta-analyses indicate a linear dose-response association with beneficial outcomes (i.e., Schwingshackl L, et al. Am J Clin Nutr. 2017;105:1462-1473 and Jayedi A, et al. Adv Nutr 2020;11:1123–1133). Concerning fish, the same EAT–Lancet Commission states: “We also suggest a range of 0–100 g/day because high intakes are associated with excellent health.” In addition to the primary benefits of fish and marine omega-3 fatty acids, inclusion of seafood in the habitual diet usually results in the displacement of other less healthy foods. So, please, do not recommend Italians to eat less fish!

Same for non-tropical oils. In the Mediterranean diet enriched with olive oil arm in PREDIMED participants consumed a mean of 50 g/day of extra-virgin olive oil and, within the context of the Mediterranean diet, this led to a 30% reduction in cardiovascular disease rates. Would you advise Italians to cut down on use of extra-virgin olive oil based on the EAT–Lancet Commission figures? On unsaturated oils, it is stated: “Evidence supports consumption of plant oils low in saturated fat as an alternative to animal fats; however, no clear upper limit of consumption exists. Thus, a wide range is suggested, and we use 50 g/day of total added fat with a mix emphasising predominately unsaturated plant oils.” If no other culinary fats are used (as customary), 50 g/d olive oil is quite close to what Italians actually consume.

The recommendation to reduce both seafood and non-tropical oils in Abstract and text should be omitted. In Table 3, the corresponding downward arrows for these foods should either be omitted or associated to an explanation in the sense of the above statements.

  1. Lines 260-262. Dietary surveys of sizable Italian cohorts have been published (refs. 34-36 and other prospective studies such as EPIC-Italy, InCHIANTI, MICOL, etc.). When data since year 2000 is available foor these studies, authors should compare them with present data, particularly on major food groups. Excessive meat consumption is probably observed in all studies.

Minor points

  1. Line 33. Revise wording.
  2. Line 51 and Table 2. Write vegetable instead of vegetal.
  3. Line 172. The CO2 equivalent of 748 kg/person/year does not correspond to the figure in Table 4 (740).
  4. Line 221-222 sentence and ref. 31 ( a self-citation) are inappropriate (see statements above).

Reviewer 2 Report

The manuscript is mainly well-written, organized, and easy to understand for the most part. This is an interesting topic with clear aims at evaluations of trends in Italic food habits and comparisons with the traditional Mediterranean diet and the targets of the EAT–Lancet Commission. However, there are some concerns which should be addressed.

  1. In the introduction section, there are many causes and effects on diet and diseases in the first and second paragraph. However, at present most evidence is uncertain or could not indicate direct causes and effects. I recommend authors to change those sentences (e.g. line 33, line 39-42) into associations between diet and diseases based on the references accordingly.
  2. In line 61, could the authors provide more details on the EAT–Lancet Commission since they have used this as the main reference. Details like what counties this guideline can be applied to, the developed countries or Italy only? Also, could authors clarify the reason of using this as a reference diet?
  3. In line 72-75, the food supply is already subtracted by the food losses, so why the authors did this again in Line 80? Could the authors provide a clearer way like a formula for clarification?
  4. Line 83, I recommend to change “food available for consumption” to “food available for human consumption” for clarification without animal feeding.
  5. In the discussion section the authors did not provide potential explanations for decreased consumption of beef meat, animal fat, milk and dairy products and increased consumption of nuts and fish. I wonder if the dietary information from the social media as invisible education could play a role since parts of scientific evidence have been propagated to the public by the modern media.
  6. Regarding explanations for reduction of fruit and vegetables consumption, do the authors think the climate changes could play a role?

Reviewer 3 Report

Thank you for the opportunity to review this manuscript. The reviewer found that the focus of this manuscript is very important. Please consider several specific comments shown below.

  1. Proportion of “food loss and waste” would change according to the progress of food processing, transportation, and preservation technology. Can you get each value in earlier and later years for Table 1? If it is expected that the proportion will not change, please add to the discussion.
  2. Table 2; Please check the unit for total energy. It will be “kcal/per capita/day”.
  3. Supplemental Table 1. Please add the unit. It will be “g/per capita/day”.
  4. It seems that there are many unnecessary spaces in the sentence. i.e. Page 2 Line 1, before “achieve”. Please check whole manuscript.
  5. Page 3 CO2 and N2O; Please change the number to subscript.
  6. References; The reviewer found many “?” characters in the references of your PDF format manuscript. Please check for garbled characters.

Round 2

Reviewer 1 Report

Authors have aswered my queries and revised the manuscript accordingly. I have no further queries.